# A Shortcut from Genome to Drug: The Employment of Bioinformatic Tools to Find New Targets for Gastric Cancer Treatment

**DOI:** 10.3390/pharmaceutics15092303

**Published:** 2023-09-12

**Authors:** Daiane M. S. Brito, Odnan G. Lima, Felipe P. Mesquita, Emerson L. da Silva, Maria E. A. de Moraes, Rommel M. R. Burbano, Raquel C. Montenegro, Pedro F. N. Souza

**Affiliations:** 1Department of Biochemistry and Molecular Biology, Federal University of Ceará, Fortaleza 60020-181, Brazil; 2Pharmacogenetics Laboratory, Drug Research and Development Center, Department of Physiology and Pharmacology, Federal University of Ceará, Fortaleza 60430-160, Brazil; 3Department of Biological Sciences, Oncology Research Center, Federal University of Pará, Belém 66073-005, Brazil; rommel@ufpa.br; 4Molecular Biology Laboratory, Ophir Loyola Hospital, Belém 66063-240, Brazil; 5Red Latinoamericana de Implementación y Validación de Guias Clinicas Farmacogenomicas (RELIVAF), Cyted, 28015 Madrid, Spain

**Keywords:** transcriptional meta-analysis, molecular docking, RT-qPCR, bioinformatics, structure-based virtual screening

## Abstract

Gastric cancer (GC) is a highly heterogeneous, complex disease and the fifth most common cancer worldwide (about 1 million cases and 784,000 deaths worldwide in 2018). GC has a poor prognosis (the 5-year survival rate is less than 20%), but there is an effort to find genes highly expressed during tumor establishment and use the related proteins as targets to find new anticancer molecules. Data were collected from the Gene Expression Omnibus (GEO) bank to obtain three dataset matrices analyzing gastric tumor tissue versus normal gastric tissue and involving microarray analysis performed using the GPL570 platform and different sources. The data were analyzed using the GEPIA tool for differential expression and KMPlot for survival analysis. For more robustness, GC data from the TCGA database were used to corroborate the analysis of data from GEO. The genes found in in silico analysis in both GEO and TCGA were confirmed in several lines of GC cells by RT-qPCR. The AlphaFold Protein Structure Database was used to find the corresponding proteins. Then, a structure-based virtual screening was performed to find molecules, and docking analysis was performed using the DockThor server. Our in silico and RT-qPCR analysis results confirmed the high expression of the *AJUBA*, *CD80* and *NOLC1* genes in GC lines. Thus, the corresponding proteins were used in SBVS analysis. There were three molecules, one molecule for each target, MCULE-2386589557-0-6, MCULE-9178344200-0-1 and MCULE-5881513100-0-29. All molecules had favorable pharmacokinetic, pharmacodynamic and toxicological properties. Molecular docking analysis revealed that the molecules interact with proteins in critical sites for their activity. Using a virtual screening approach, a molecular docking study was performed for proteins encoded by genes that play important roles in cellular functions for carcinogenesis. Combining a systematic collection of public microarray data with a comparative meta-profiling, RT-qPCR, SBVS and molecular docking analysis provided a suitable approach for finding genes involved in GC and working with the corresponding proteins to search for new molecules with anticancer properties.

## 1. Introduction

Gastric cancer (GC) is a complex and multifactorial disease caused by genetic, epigenetic and environmental influences [1]. Although a worldwide decline in incidence and mortality rates has been observed in recent decades, GC still constitutes a significant burden and significantly impacts populations in developing countries. Current statistics reveal that GC is the third most common malignancy and the fourth leading cause of cancer-related mortality worldwide, accounting for more than 768,000 deaths in 2020 [2,3]. The National Cancer Institute (INCA) estimates 13,340 new cases of GC in men and 8140 in women for Brazil in the three years from 2023 to 2025. These values correspond to an estimated risk of 12.63 new cases per 100,000 men and 7.36 per 100,000 women [4].

Most gastric cancers are adenocarcinomas. According to Lauren’s classification, these represent more than 95% of all gastric malignancies and can be subdivided into intestinal and diffuse types. This classification is based on the histology of the tumor [5]. The intestinal type consists of differentiated cancer with a tendency to form glands. It progresses mainly through successive changes in the normal gastric mucosa, leading to acute and chronic gastritis, atrophic gastritis, intestinal metaplasia, dysplasia and a gastric tumor [5].

The standard treatment for gastric cancer is based on the triad of surgery, chemotherapy and radiotherapy. Surgical resection is considered the primary method of treatment at an early stage and the only potentially curative approach in treating gastric cancer. However, recurrence is frequently observed in many patients, even after resection. Faced with this problem, a study of 206 patients with gastric cancer revealed that patients in stages II and III had better survival rates with adjuvant chemotherapy than surgery alone [6]. Over two decades ago, Phase II clinical studies showed that neoadjuvant chemotherapy can increase tumor resection success rate by 72 to 87% [7,8].

Various chemotherapeutic drugs and therapeutic schemes are approved for the pharmacological treatment of gastric cancer. Some of these therapeutic schemes are well described in the literature: the FLOT scheme (5-fluorouracil, leucovorin, oxaliplatin and taxane), the ECF scheme (epirubicin, cisplatin and fluorouracil), FOLFOX (5-fluorouracil, leucovorin and oxaliplatin), infusional cisplatin (CF) and a single-drug regimen with irinotecan [9,10].

Nowadays, two problems related to drugs for GC need to be solved: (1) the high toxicity of drugs to patients and (2) the resistance of GC cells to these drugs. In relation to these problems, bioinformatics has been raised as a powerful tool to identify new targets or new molecules for a target. Over the last few years, bioinformatics has helped researchers worldwide to rapidly find molecules and targets that could be applied to cancer. For example, Miller et al. [11] performed a structure-based virtual screening (SBVS) to find new molecules with activity against the proteosome of pancreatic cancer cells. In the study, the authors analyzed 380,000 compounds against one proteasome subunit. After SBVS analysis, out of 288 compounds tested in vitro, 1 was selected for experiment analysis [11]. This case is an example of how bioinformatics can rapidly help drug discovery.

Another study by Rohr et al. [12] developed a pipeline employing R package scripts to analyze data from transcriptional analysis from a microarray from pre- and post-malignant colorectal cancer. Using the pipeline developed, it was possible to analyze a merged dataset containing 231 normal, 132 adenoma and 342 colon cancer tissue samples across twelve independent studies analyzed by microarray deposited in the Gene Expression Omnibus (GEO) [12]. The analysis was performed to understand the genes involved in the disease severity, progression, and new targets for treating colorectal cancer. Both studies employed bioinformatics to find new targets and molecules that could be useful in cancer treatment, corroborating bioinformatics applications in clinics.

The present study aimed to identify new potential targets involved in GC establishment in samples collected from different populations worldwide using bioinformatic tools. First, an R code script was employed to analyze the microarray metadata from healthy patients and with GC downloaded from GEO in a search for new genes involved in GC establishment.

## 2. Methodology

### 2.1. Selection of Microarray Transcriptome Studies

To minimize the number and impact of batch effects, Minimum Information About a Microarray Experiment (MIAME)-compliant microarray studies were selected based on predefined inclusion criteria: (1) collection of human tissue samples; (2) use of the GPL570 platform (Affymetrix Human Genome U133 Plus 2.0 array) for matching probe sets (Appendix A); and (3) inclusion of samples from gastric cancer (tumor) and healthy tissue (normal). In total, 3 independent studies were chosen from an initial list of 156 to construct the meta-data.

### 2.2. Data Acquisition and Processing

Data acquisition and processing were performed in the R studio (v 4.1.3). Raw data were downloaded from the GEO database (https://www.ncbi.nlm.nih.gov/geo/) (accessed on 10 January 2022) using the getGEOSuppFiles function of the GEOquery package (version 2.58.0) and the method described by Rorh et al. [12]. The .cel format files for each study were unpacked using the R untar function and loaded using the ReadAffy function of the affy package (version 1.68.0). Subsequently, the data were corrected and log-transformed by the frozen Robust Multiarray Averaging (fRMA) method using the fRMA package (version 1.42.0). Compared to traditional RMA, fRMA uses pre-calculated probe drifts to normalize the raw microarray data and has been shown to outperform RMA when preprocessing individual data sets for pooled analyses [12].

### 2.3. Metadata Construction

The construction of the metadata followed the script described and validated by Rohr and colleagues [12]. Following fRMA normalization, the matrices of individual datasets were merged by combining the microarray probes. The batch effect among studies was identified using the Uniform Manifold Approximation and Projection (UMAP) method using the umap package (version 0.2.7.0) and removed using the original ComBat parametric iteration within the SVA package (version 3.38.0). UMAP was used over traditional principal component analysis (PCA) to identify batch effects due to its ability to represent local relationships better while preserving the global structure [12]. The ComBat function was chosen due to its flexibility, reliability, and ability to define covariates of interest.

Next, probes with an expression variation in the 75th percentile were filtered from the meta-dataset using the gene-filter function in the oligo package (version 1.54.1). Finally, the maximum average expression using the hgu133plus2 collapsed redundant probes to their corresponding human gene symbol.db package (version 3.2.3). The constructed metadata contain 601 samples, including 152 tumor and 449 normal patients from 3 independent studies.

Differential expression (DE) analysis for tumor versus normal comparison was performed using the limma package. Specifically, DE analysis was performed independently on 25% of the most variable genes between each group. Genes were considered differentially expressed if the q-value of |LogFC| ≥ 1.0 and false discovery rate (FDR) < 0.01.

### 2.4. Gene Set Enrichment Analysis (GSEA)

The significantly enriched tumor and normal phenotype gene sets were identified using Gene Set Enrichment Analysis (GSEA, gsea-msigdb.org) (accessed on 15 January 2022). The H collection (hallmark gene set) available in the Molecular Signatures Database (MSigDB) 3.0 was chosen for the enrichment analysis [13]. The default parameters defined by Subramanian et al. [14] were used in this analysis. One hundred permutations determined the significance of the GSEA analysis (corrected *p*-value < 0.05 and false discovery rate (FDR) < 0.25). GraphPad Prism™ software version 5 (San Francisco, CA, USA) was also used to represent individual genes’ enrichment scores and signal-to-noise values.

### 2.5. Differential Expression and Survival Analysis from TCGA

Based on tumor and normal samples from the TCGA (The Cancer Genome Atlas) databases, differential expression and overall survival analyses were performed for the three up- and downregulated genes in the differential expression analysis using the limma package and for the three most enriched genes in the analysis using the GSEA. Using the Gene Expression Profiling Interactive Analysis (GEPIA) tool, available at http://gepia.cancer-pku.cn/ (accessed on 19 August 2023), an analysis of the differential expression of the mRNA levels of these proteins in gastric cancer (identified by STAD) was performed. The results are presented in box plots. The Kaplan–Meier plotter database (https://kmplot.com/analysis/) (accessed on 21 August 2023) was used to compare the expression profiles of the gene of interest in normal and tumor gastric tissue, classifying them as high or low/medium expression, which was then correlated with the probability of patient survival [15]. This tool can evaluate the correlation between the expression of all genes (mRNA, miRNA, protein) and survival in more than thirty thousand samples of different types of tumors. Therefore, the genes most significantly associated with patient survival can be identified.

### 2.6. Primer Designer

The Thermofisher trading platform (https://apps.thermofisher.com/apps/oligoperfect/#!/design) (accessed on 15 March 2022) was chosen to design the perfect primers for polymerase chain reaction (PCR). For this, the mRNA codes had to be obtained from the NCBI databases (https://www.ncbi.nlm.nih.gov/) (accessed on 18 March 2022), providing the nucleotide sequence for all transcript variants of interest (Appendix A). The primer sequence was provided by the OligoAnalyzer™ Tool from Integrated DNA Technologies.

### 2.7. Cell Culture

The gastric cancer cell line AGP-01 was obtained from the malignant ascitic fluid of primary metastatic intestinal-type tumors. The ACP-02 and ACP-03 cell lines were established from Brazilian patients’ diffuse-type and primary intestinal-type tumors, respectively. The AGS cell line was obtained from gastric adenocarcinoma tumors of a Caucasian patient [16,17]. The non-malignant gastric cell lineage MNP-01 was previously established from normal gastric mucosa and was used as a cell-type-specific control [18]. All cells were cultured in Dulbecco’s modified Eagle’s medium (DMEM; Gibco^®^), supplemented with 10% (*v*/*v*) fetal bovine serum (Gibco^®^), 1% (*v*/*v*) penicillin (100 U mL^−1^) and streptomycin (100 mg mL^−1^) (Invitrogen^®^), in a humidified atmosphere with 5% CO_2_ at 37 °C. Cell confluence was observed under a conventional microscope.

### 2.8. Gene Expression by qRT-PCR

#### 2.8.1. RNA Extraction

Total mRNA from gastric cell lines and tissue specimens was extracted using TRIzol™ reagent (Invitrogen, Carlsbad, CA, USA) as per the manufacturer’s guidelines. Quantification was performed in a spectrophotometer (NanoDrop^®^, Thermo Scientific, Carlsbad, CA, USA), and mRNA was stored at −80 °C until conversion into cDNA.

#### 2.8.2. Conversion of mRNA to cDNA

The mRNA was converted to cDNA using the High-Capacity cDNA Reverse Transcriptase^®^ kit (Thermo Scientific, Carlsbad, CA, USA). A thermocycler was used to carry out the conversion reactions (Verity^®^, Thermo Scientific, Carlsbad, CA, USA).

#### 2.8.3. Real-Time Quantitative PCR (qRT-PCR)

Gene expression was determined by qRT-PCR using PowerUp SYBR^®^ Master Mix (Life Technologies, San Diego, CA, USA) on a QuantStudio5 Real-Time PCR system (Applied Biosystems^®^, Carlsbad, CA, USA). The primer efficiency was determined for all described genes. The relative expression levels of the *AJUBA*, *CD80*, *NOLC1* and *KNL1* genes were normalized and determined using the ACTB gene as an endogenous control. Calculations were performed using the 2^−∆∆CT^ method. The requirements proposed in the Minimum Information for Publication of Quantitative Real-Time PCR Experiments (MIQE) Guidelines were followed [19].

### 2.9. Three-Dimensional Structure Obtention

The 3D structures of the proteins were obtained from the AlphaFold Protein Structure Database (https://alphafold.com/) (accessed on 15 June 2022), an artificial intelligence (AI) system developed by DeepMind and EMBL-EBI that predicts the 3D structure of proteins from an amino acid sequence with high precision. The predictions of the 3D structures related to the *AJUBA*, *CD80* and *NOLC1* genes were available in this database free of charge and with open access [20,21]. Even though they were produced by artificial intelligence, the structures had the quality assessed by Verify 3D and ERRATA servers.

The organism filter was applied to choose the structure for each protein, selecting the *Homo sapiens* option. Per-residue confidence score (pLDDT) and FASTA amino acid sequence coverage analysis were considered. The PyMOL Molecular Graphics System (version 2.5.4, Schrödinger, LLC, San Diego, CA, USA) [22] was employed to analyze the 3D structures of proteins. The 3D structures of the *AJUBA*, *CD80* and *NOLC1* proteins downloaded from the AlphaFold Protein Structure Database (https://alphafold.com/) (accessed on 25 June 2022) have the following Uniprot codes Q96IF1, P33681 and Q96J17, respectively [20,21].

### 2.10. Virtual Screening

To identify inhibitors of *AJUBA*, *CD80* and *NOLC1* proteins, a structure-based virtual screening (SBVS) analysis was performed on the Mcule online platform (https://mcule.com/dashboard/) (accessed on 10 August 2022). Mcule is a platform that provides information technology (IT) infrastructure containing drug discovery tools, a high-quality compound database, pharmacokinetic and toxicological prediction tools, and commercialization services for approximately 100+ million compounds [23,24].

On this platform, virtual screening workflows consisting of a set of filters and calculations can be created. Filters can remove compounds that are unlikely to bind to study targets or have undesirable properties. The calculations, in turn, rank the best candidates concerning the binding affinity between the protein and ligand complex. Additionally, was is possible to perform Hit Identification analyses, and then the Structure-Based Virtual Screen option was selected, generating the workflow. In the collection, Mcule Purchasable (Full) was selected, aiming to guarantee that the evaluated molecules would be available for commercialization. In the basic property filter, only the RO5 violation filter was changed, with an analysis performed with the maximum value of violations equal to 0 (RO5 violations 0) for each target. The other default settings preserved, such as sampling and diversity filters, were used to randomly select several different chemical structures with a maximum value of 10 rotating bonds, 5 chiral centers and 0 or 1 violation of Lipinski’s rule of five.

Lipinski’s rule of five allows a prediction of the oral bioavailability profile for new molecules. In this rule, it is established that for a compound to be a good drug candidate, it must present multiple values of 5 for the following 4 parameters: (1) log P greater than or equal to 5; (2) molecular mass less than or equal to 500; (3) number of hydrogen bond acceptors less than or equal to 10; (4) number of hydrogen bond donors less than or equal to 5. It is defined that a molecule may present only one violation of one of these parameters to be a promising drug candidate [25,26].

In summary, using the Mcule platform, it was possible to identify compounds that bind to the active or allosteric site of the targets under study, obtain results on the binding affinity of the protein–ligand complex (docking scores) and conduct an analysis of the toxicity of compounds from their chemical structure.

### 2.11. In Silico Toxicity Prediction

The top ten compounds with minimum target binding energy were selected in each virtual screening analysis performed in Mcule for the target proteins *AJUBA*, *CD80* and *NOLC1*. Toxicological parameters were estimated for these compounds using Mcule and ProTox-II.

On the Mcule platform, it is only possible to predict the toxicity of molecules contained in its compound database. This in silico prediction is based on the search for substructures commonly found in toxic and promiscuous ligands. This way, when a molecule is classified as toxic, the responsible fragment is flagged and identified [23]. All molecules analyzed in this tool were also evaluated in the ProTox-II server.

ProTox-II, freely available from https://tox-new.charite.de/protox_II/ (accessed on 4 September 2022), is a web server for in silico toxicity prediction incorporating molecular similarity, pharmacophores, fragment propensities and machine learning models for the prediction of toxicity endpoints, specifically acute toxicity, hepatotoxicity, cytotoxicity, carcinogenicity, mutagenicity, immunotoxicity, pathways of adverse outcomes (Tox21) and toxicity targets. It is worth mentioning that the predictive models of this server are built based on data from in vitro trials and in vivo cases [27].

For evaluating the toxicity profile in ProTox-II, the SMILE code was used as an input file for the construction of the two-dimensional chemical structure of the chemical products. Acute oral toxicity was determined using the toxicity class. Only compounds that showed prediction for toxicity class equal to or greater than IV and no prediction for toxicity endpoint models were selected for the next step.

### 2.12. Prediction of ADME Parameters

To evaluate pharmacokinetic parameters, the SwissADME tool was used. The SwissADME tool assesses the pharmacokinetics and structural similarity between molecules. This tool also allows for calculating physicochemical descriptors. It provides access to robust and fast predictive models for physicochemical properties, similarity to drugs and compatibility with medicinal chemistry [28,29].

### 2.13. Molecular Docking (MD) Assays

Only chemicals obtained after virtual screening that showed favorable toxicological properties (prediction for toxicity class equal to or greater than IV and no prediction for toxicity endpoint models) in the in silico toxicity prediction step were selected and virtually tested for potential binding against study targets using the DockThor server. This server presents a methodology using flexible ligands and rigid receptors through a genetic algorithm and the MMFF94s molecular force field to predict the score of each pose [30,31].

For each analysis, the 3D structure of the protein was loaded as a PDB file into the docking platform. In the protein preparation step, the internal program PdbThorBox automatically parameterizes the protein atoms according to the atomic type and partial charges of the MMFF94 force field, adding polar hydrogen atoms when necessary, reconstructing missing atoms from residual side chains and adjusting the protonation state. In the ligand preparation step, the internal MMFFLigand program parameterizes the atoms according to the atomic type and partial charges of the MMFF94s force field, adding polar hydrogen atoms if required by the user.

In the docking step, to find the protein cavities and the site of interaction between the molecules, blind docking was performed, with an option called Blind Docking. This option generates a grid box centered on the protein coordinate center with a size that covers the entire receptor. The prediction of affinity and total energy of the protein–ligand complexes was performed using the internal program DockTScore. At the end of the molecular docking assay, the final output compounds were ranked based on their minimum target binding energy. The PyMOL Molecular Graphics System (version 2.5.4, Schrödinger, LLC, San Diego, CA, USA) [22] and BIOVIA Discovery Studio Visualizer (version 21.1.0.20298) tools were used to analyze the docking conformation of the protein–ligand complex and the types of connection involved.

### 2.14. Protein–Protein Interaction Network

To identify the protein interaction network of the *AJUBA*, *CD80* and *NOLC1* proteins, the STRING tool (version 11.5) was used. The STRING resource is available online at https://string-db.org/ (accessed on 5 January 2023). The results obtained using this tool enabled the selection of protein complexes that later served as controls for the molecular docking (MD) assay.

### 2.15. Re-Docking

For the re-docking process, it was necessary, initially, to select a protein–protein complex from the interaction network provided by the STRING tool. Priority was given to choosing known complexes with comfort from curated databases and experimentally determined. Therefore, molecular docking experiments were performed using the ClusPro 2.0 server (https://cluspro.org) (accessed on 20 January 2023), the best-performing server currently available for the CAPRI challenge [32]. The GPU option was selected because it uses more specific computer graphics units from the Massachusetts Green High-Performance Computing Center (MGHPC). The best complexes generated by molecular docking studies were analyzed regarding interface energy and interaction between residues. This analysis sought to evaluate and understand how the proteins interact with each other, observing the predominant interaction pose and the minimum energy value.

### 2.16. Statistical Analysis

Data were shown as a mean ± standard deviation (SD) and the groups were compared with each other by Analysis of Variance (ANOVA) followed by Bonferroni’s posttest. Significant differences were considered with an interval of confidence of 95% (*p* < 0.05). GraphPad Prism 5.01 (San Francisco, CA, USA) software was used for data analysis and graph design.

## 3. Results

### 3.1. Differential Gene Expression in Gastric Tumor Meta-Dataset

We constructed a meta-dataset for transcriptome analysis to assess the gene expression profile of gastric cancer tissue versus normal gastric tissue. After batch correction among datasets (Figure 1A), the differentially expressed genes were demonstrated in a volcano plot, representing the significance of the adjusted *p*-value and fold change (Figure 1B). Thus, the top ten genes with the highest and lowest values of fold change (log), whose *p*-values were significant (*p* < 0.05), were organized, as shown in Table 1. The highest gene expression levels in the gastric tumor dataset were observed for *AJUBA* (1.44 × 10^14^), *GPNMB* (1.11 × 10^14^) and *CD80* (1.09 × 10^14^). Otherwise, the lowest expression levels were observed for the *FBXL13* (−1.56 × 10^14^), *PDILT* (−1.41 × 10^14^) and *CCDC69* (−1.28 × 10^14^) genes. Those up- and downregulated genes were chosen because our focus was to develop a methodology effective for identifying gene markers in GC development. If we compare control and cancer cells, the top three up- and downregulated genes were most associated with cancer development (upregulated genes) or not (downregulated genes) (Table 1). Therefore, it is feasible to think that the genes overexpressed in cancer cells are important for cancer development. Based on that, they are good targets for developing new target-directed drugs.

### 3.2. Gastric Tumor Group Involved in Multiple Cancer Progression Pathways

In addition to the analysis with R Studio (Table 1), the GSEA platform was also used to determine gene enrichment in gastric tumor samples [14]. For this, the GSEA provided graphs with the specifications of the 18 main hallmarks in the tumor and normal samples. GSEA assessed the functional differences by comparing gastric tumors versus normal gastric tissue. For instance, gastric tumors were positively correlated with several cancer hallmarks (Figure 2). The most enriched pathways were the G2M checkpoint (NES = 2.22, *p* = 0.00001), spermatogenesis (NES = 2.00, *p* = 0.00001) and E2F targets (NES = 2.05, *p* = 0.00001). Therefore, an intensive regulatory role was observed for the development and progression of gastric tumors, exhibiting significant changes in pathways.

### 3.3. Expression and Prognostic Value of Genes Involved in Gastric Cancer Progression

In the analysis of gene expression associated with prognosis, nine genes were selected. Three genes were overexpressed (*AJUBA*, *GPNMB* and *CD80*), three were underexpressed (*FBXL13*, *PDILT*, *CCDC69*) and three other genes were enriched (KNL1, IL13RA and *NOLC1*). Comparison between the mRNA expression levels of these genes in gastric cancer (identified as STAD) and in normal tissues revealed the following results with statistical significance (*p* < 0.05): overexpression of the *AJUBA*, *CD80*, *NOLC1* and KNL1 genes and hypoexpression of the CCDC69 gene in gastric cells (Figure 3). The results of the evaluation of whether the most relevant genes were associated with poor survival in gastric cancer patients are shown in Figure 4. Interestingly, the expression levels of seven genes were significantly correlated with the survival of patients with gastric cancer. Prognostic analysis using the Kaplan–Meier plotter revealed that the high expression of *AJUBA* (*p*-value = 6.1 × 10^−6^), *CD80* (*p*-value = 1.1 × 10^−8^), CCDC69 (*p*-value = 3.3 × 10^−3^), *KNL1* (*p*-value = 0.00034) and *NOLC1* (*p*-value = 6.9 × 10^−3^) and the low expression of *GPNMB* (*p*-value = 0.0033) and *PDILT* (*p*-value = 0.031) were significantly correlated with poor prognosis in patients with gastric cancer. The overexpressed genes associated with a worse prognosis were selected for (1) the design of their primers and, subsequently, (2) undergoing polymerase chain reaction (PCR) for validation in a gastric cancer cell line model.

### 3.4. Validation of Relevant Genes in Gastric Tumor Cell Lines Compared to Normal Gastric Cell Lines

To validate and evaluate how these selected genes are expressed in gastric cancer in an in vitro model, we compared the relative gene expression levels of gastric cancer cell lines (ACP-02, ACP-03, AGP-01 and AGS) and a non-malignant gastric cell line (MNP-01). The results showed a significant reduction in *KNL1* gene expression in ACP-02 cells compared to MNP-01 (Figure 5A). In the case of ACP-03, *AJUBA*, *CD80*, *NOLC1* and *KNL1* were overexpressed when compared to MNP-01 (Figure 5B). For the AGP-01 cell line, *AJUBA* in AGP-01 presented an increased expression compared to MNP-01. In contrast, *KNL1* presented a reduced expression (Figure 5C). For the AGS cell line, *AJUBA* and *KNL1* genes presented higher expression compared to MNP-01 (Figure 5D).

### 3.5. In Silico High-Throughput Virtual Screening

Using a structure-based approach to drug screening, an extensive collection of small molecules was investigated for their potential to interact with a designated target protein. In this investigation, a screening process involving 10,000 ligands from the MCULE library was conducted. The selection of target proteins was guided by overexpression data obtained from gastric cancer cell lines, specifically focusing on genes such as *AJUBA*, *CD80* and *NOLC1*. The KNL1 protein was not selected for further analysis of SBVS because its mRNA expression by RT-qPCR showed upexpression in two lines of GC cells and downexpression in the other two lines of GC cells. Based on these controversial results, it was better to move forward only with three genes selected: *AJUBA*, *CD80* and *NOLC1*. Through virtual screening, the top 100 ligands were initially identified and subsequently refined to the top 10 based on VINA scores (Appendix A). These chosen ligands were further evaluated. The molecules found were suitable for subsequent analysis, considering their favorable interaction energies and toxicological characteristics.

### 3.6. Molecular Docking (MD) Validation

The in silico evaluation of the best hits at the end of each virtual screening, selecting a compound with the best prediction of pharmacological parameters for each protein target studied, was possible. The molecules were designated with Mcule IDs as MCULE-2386589557-0-6, MCULE-9178344200-0-1 and MCULE-5881513100-0-29. The best molecules were chosen based on (1) the best pose of interaction (a site important for protein activity), (2) the number of interactions, (3) the distance of bonds and (4) toxicity (Appendix A). The formula, chemical structure and molecular weight of these chosen compounds are shown in Table 2.

Molecular docking analyses revealed that the compound MCULE-2386589557-0-6 presented a polar bond interaction with the Val^443^ and two pi-alkyl interactions with the Leu^235^ and Pro^232^ residues of *AJUBA* (Figure 6A). MCULE-9178344200-0-1 formed two polar bonds with the Glu^177^ and Asp^205^ residues, one Alkyl with Ile^185^ and one pi-alkyl Leu^182^ with the *CD80* protein (Figure 6B). Regarding the compound MCULE-5881513100-0-29 and the *NOLC1* protein, a polar interaction with Arg^390^ and a pi–cation interaction with the residue Lys387 (Figure 6C) were identified.

Concerning docking scores for the best poses, Table 3 shows the minimum binding energy for each receptor–ligand complex evaluated and the prediction of the toxicity parameters LD50 value and acute oral toxicity, determined by the middle of the toxicity class. All three selected compounds show a prediction for toxicity class equal to or greater than IV and no prediction for toxicity endpoint models. It is essential to consider that on the toxicity scale, class I represents the compounds most likely to manifest acute oral toxicity, and class VI represents the compounds with higher safety scores.

### 3.7. In Silico Pharmacokinetics Prediction

Table 4 indicates the results related to the ADME in silico parameters of the three selected compounds: (1) MCULE-2386589557-0-6, (2) MCULE-9178344200-0-1 and (3) MCULE-5881513100-0-29. Pharmacokinetic data were obtained from the SwissADME platform and included information on molecular weight (PM), topological polar surface area (TPSA), lipophilicity coefficient (cLogP), solubility in aqueous media (LogS), gastrointestinal absorption (AGi), penetration of the blood–brain barrier (BBB) and data on P-glycoprotein (P-Gp) substrates.

As shown in Table 4, the compound with the highest coefficient of lipophilicity is MCULE-2386589557-0-6 (cLogP = 3.81). Consequently, this compound has a lower solubility value in aqueous media (LogS = −4.75). All analyzed compounds presented satisfactory TPSA. This parameter is used in medicinal chemistry to describe the ability of a molecule to permeate cells and physiological barriers [29]. This data comprise the basis for discussing possible pharmacokinetic properties and the analysis of experimental assays since drug transport is required to reach the site of pharmacological action.

All compounds analyzed showed high absorption via the gastrointestinal tract (Table 4). Regarding penetration through the blood–brain barrier, only compounds (1) MCULE-2386589557-0-6 and (3) MCULE-5881513100-0-29 showed permeability capacity. The compound (3) MCULE-5881513100-0-29 was classified as a P-glycoprotein (P-Gp) substrate. Expressed in cell membranes, P-Gp acts as an efflux pump of some substances, especially xenobiotics, from the interior of cells.

### 3.8. Protein–Protein Interaction Network

By analyzing the network of protein–protein interactions (PPIs), it was possible to evaluate the pathways of our targets (Figure 7); somehow, the molecules chosen to interfere with those pathways (Figure 8). The analysis performed on the STRING database (Figure 7) revealed that *AJUBA*, *CD80* and *NOLC1* are involved in a complex pathway with many other proteins. Based on those analyses, one protein for each target was selected following the score provided by STRING to perform a re-docking analysis to assess our molecules’ interference in PPI interaction molecules. For *AJUBA* (Figure 8, Panel 1), the protein selected was CTNNB1; for *CD80* (Figure 8, Panel 1), the protein selected was CTLA4; and for *NOLC1* (Figure 8, Panel 1), the protein selected was NHP2. In the re-docking analysis, the control was the interaction between the proteins without our molecules (Figure 8, Panel 2). After the re-docking analysis of our targets complexed with each proposed molecule and the proteins selected, it was clear that interaction happens in a different site compared to the control (Figure 8, Panel 3). These results strongly suggest that the PPI is affected by the presence of molecules selected, which could indicate an interference in the pathway.

## 4. Discussion

Despite advances in science and the various therapeutic options currently available for controlling gastric cancer, the prognosis of patients affected by it remains unfavorable. In this context, chemotherapy resistance and the serious adverse effects of conventional pharmacological treatments represent a major obstacle to the successful treatment of gastric cancer.

Drug resistance is a phenomenon that results from a variety of pharmacokinetic and molecular changes. It refers to the ability of microorganisms or cancer cells to resist the effects of a normally effective drug against them. Although many types of cancer are initially susceptible to chemotherapy, with prolonged use, the development of resistance to chemotherapy drugs is observed through different mechanisms, such as inactivation or reduction in drug activation; changing drug targets; drug efflux; DNA damage, repair and metabolic changes; inhibition of cell death; epithelial–mesenchymal transition and metastasis; heterogeneity of cancer cells; and epigenetic modifications, with the possibility of combining any of these mechanisms [33].

As a new type of approach, targeted therapies play an important role in the treatment of gastric cancer by interfering with gene expression or target proteins that play or regulate critical roles in tumor growth or progression, providing the new generation of chemotherapy drugs for the treatment of cancer with greater selectivity and efficacy and reduced toxicity [34]. Genomic and genetic studies provide valuable information about genomic changes in tumor samples compared to normal tissue samples. These studies provide important tools for understanding key information about tumor initiation, progression and metastasis [35]. As shown in the present study, evaluating the transcriptional profile in gastric tumor samples is a strong strategy for identifying new molecular targets.

In searching for drug targets and discovering new drug candidates, computer-aided drug design (CADD) methodologies stand out as a powerful and promising technology for faster, cheaper and more effective drug design in drug research [36]. Furthermore, they provide an absolute starting point for drug target discovery. Faced with the high cost and time required for research and development of drugs in the oncology area, computational models represent an efficient alternative since they are capable of predicting physical–chemical, pharmacokinetic, pharmacodynamic and toxicological parameters from a given molecular structure, as well as optimizing the in vitro test step [36].

The results obtained so far in obtaining new drugs through in silico studies are well known. For example, there is the development of the drug captopril, the first angiotensin-converting enzyme (ACE) inhibitor and one of the first successful drugs produced using computational tools to optimize drug planning in the 1980s [37]. Following this study, structure-based drug development exhibited a significant impact on drug design with an increasing number of applications, and a rapid growth of computational tools for drug discovery, including anticancer therapies, was observed.

Structure-based virtual screening (SBVS) is a robust technique that allows rapid identification of biologically active compounds, providing an efficient and cost-effective alternative to high-throughput experimental screenings. This technique allows the prediction of the best mode of interaction between two molecules to form a stable complex. It uses scoring functions to estimate the strength of a non-covalent interaction between a ligand and molecular target. Different successful examples of SBVS applications are reported in the literature, evidencing the versatility, high performance and great utility of SBVS in drug discovery programs. In this study, the SBVS technique enabled the identification of synthetic molecules with great potential to inhibit target proteins relevant to carcinogenesis and the establishment of gastric cancer. The ligands with the best classification, considering the minimum energy of binding to the target, had their ADMET properties evaluated and underwent a molecular docking validation protocol.

Analyzing ADMET properties is an important step in drug design development. ADMET refers to the processes of absorption (A), distribution (D), metabolism (M), excretion (E) and toxicity (T) still in the early stages of the drug discovery process. This step drastically reduces the fraction of failure related to pharmacokinetics in the clinical phases and the toxic effects associated with drugs [28].

Faced with the need to develop new therapeutic options to improve the effectiveness of gastric cancer treatment and patient survival statistics, targeted therapies stand out as instruments with great potential for therapeutic success. In this context, the present study points to the *AJUBA*, *CD80* and *NOLC1* proteins as potential candidates for targeted therapy in the treatment of GC.

The *AJUBA* protein has been implicated in the development of several human cancers. It is known that this protein participates in the assembly of countless protein complexes and is involved in several cellular biological processes, such as the repression of gene transcription, cell–cell adhesion, mitosis, differentiation, proliferation and cell migration [38]. Previous studies have shown that the *AJUBA* protein promotes colorectal cancer cell growth by suppressing the JAK1/STAT1/IFIT2 network and activating N-cadherin expression through interaction with Twist in colorectal cancer cells [39,40]. However, its expression pattern and biological significance in gastric cancer are still not fully elucidated.

In the present study, analysis of the microarray metadata revealed that *AJUBA* gene expression was higher in gastric cancer samples than in normal tissues. By comparing the relative expression levels of this gene in samples of gastric tumor cell lines (ACP-02, ACP-03, AGP-01 and AGS) and normal gastric cells (MNP-01) using qRT-PCR, the results confirmed a significant increase in the *AJUBA* gene expression level in AGP-01 and AGS cell lines. Data from the survival analysis obtained using the Kaplan–Meier plotter database revealed that the increase in *AJUBA* expression is closely associated with a reduction in the overall survival rates of patients affected by GC.

Dommann et al. [41] developed a study that analyzed the transcriptome of SW480 human colon cancer cell lines by RNA sequencing and confirmed the sequencing data with biological assays. In this analysis, it was possible to conclude that cells devoid of *AJUBA* were less proliferative, more sensitive to irradiation, migrated less and were less efficient in forming colonies. Furthermore, loss of *AJUBA* expression decreased tumor burden in a murine model of colorectal metastasis to the liver [41].

From these data, it is assumed that the inhibition of the *AJUBA* protein in patients with gastric cancer can reproduce the phenomena observed in the work of Dommann et al. [41]. Thus, compounds that inhibit this target may represent a new therapeutic alternative for GC.

*CD80* plays an important role in T-cell activation, exerting a dual effect on tumor immunity: it binds to CD28 to provide a costimulatory signal for T-cell activation, and it binds to CTLA-4, resulting in an immunosuppressive effect. It is understood that the binding of *CD80* with CTLA-4, a receptor that acts as an important negative regulator of T-cell responses, is favorable for carcinogenesis because cancer cells commonly use the immunosuppressive function of regulatory T cells to avoid immunological attacks [42,43,44]. 

Based on this understanding, a monoclonal antibody against CTLA-4 (ipilimumab) was approved by the Food and Drug Administration (FDA) for the treatment of melanoma [45]. Ipilimumab blocks the co-inhibitory signal induced by CTLA-4 binding with *CD80* to enable CTL-mediated antitumor immunity [44].

Considering the positive results of blockade of the co-inhibitory signal induced by the binding of CTLA-4 with *CD80* for patients with melanoma, it is believed that it is important to obtain a greater understanding of expression data and the function of *CD80* in gastric cancer, as well as to characterize the existing molecular interaction between the CTLA-4 and *CD80* proteins.

The analysis of the protein interaction network developed in the present study using the STRING v11.5 tool made it possible to evaluate the protein–protein interaction network. Among the results of our study, it was revealed through a molecular docking assay that the molecular binding between the CTLA-4 and *CD80* proteins is altered after the anchoring of the MCULE-9178344200-0-1 compound to the *CD80* protein (Figure 6). Alteration of the molecular binding between the two proteins may result in the functional change of this interaction and, consequently, promote the blockade of the co-inhibitory signal induced by the binding of CTLA-4 with *CD80*.

Concerning *CD80* expression and function data in gastric cancer, the results of our study revealed an increase in the expression level of the *CD80* gene in samples of gastric cancer tumors compared to normal tissues in the microarray metadata analysis, as well as in real-time PCR detection, which revealed a significant increase in the expression level of the *CD80* gene in the ACP-03 cell line. Survival analysis data obtained using the Kaplan–Meier plotter database revealed that increased *CD80* expression is associated with reduced overall survival rates of patients affected by GC. These results show the protein and the *CD80* gene as potential successful therapeutic targets for the control of GC.

On the other hand, in a study by Feng et al. [44], the determination of *CD80* mRNA levels in gastric adenocarcinoma tissues and adjacent normal tissues by RT-qPCR was performed. As a result, it was seen that *CD80* is downregulated in gastric cancer tissues in 15 out of 20 patients compared to normal gastric tissue. Specifically, 70% of gastric tumor tissues demonstrated reduced *CD80* expression [44]. Thus, it is important to consider that the *CD80* gene is differentially expressed in gastric adenocarcinoma cell lines. 

The nucleolar and coiled-body phosphoprotein 1 (*NOLC1*) protein is responsible for various cellular life activities, including ribosome biosynthesis, DNA replication, transcription regulation, RNA processing, cell cycle regulation, apoptosis and cell regeneration [46]. Our results demonstrated that gene expression was higher in samples of gastric cancer tumors than in normal tissues (Figure 2), which was confirmed by RT-qPCR for the GC cell lines ACP-03 and AGS (Figure 4). Data obtained using the Kaplan–Meier plotter database revealed that patients affected by GC with overexpression of *NOLC1* have shorter overall survival than those with low expression of *NOLC1* (Figure 3). Additionally, the gene enrichment analysis suggests the participation of the *NOLC1* gene as an important regulator for the development and progression of gastric tumors.

In a study carried out by Kong et al. [47], the role of *NOLC1* in esophageal cancer (ESCA) was determined, and its gene expression in ESCA tissues and cell lines was evaluated by qRT-PCR, immunohistochemistry or Western blotting. Among the results, overexpression of *NOLC1* was observed in ESCA tissues and ESCA cell lines (EC9706, Eca109, TE-13, Kyse170, T.TN) compared to adjacent normal tissues and normal esophageal cell lines. *NOLC1* overexpression was markedly associated with larger tumor size, lymph node metastases and advanced TNM stage.

The results of the correlation between *NOLC1* gene expression and overall survival using Kaplan–Meier plotter for patients with esophageal cancer coincided with the data obtained for patients with gastric cancer, evaluated in our study. So, *NOLC1* overexpression was also associated with reduced overall survival rates for patients with ESCA. *NOLC1* knockdown, in turn, inhibited proliferation, migration, invasion and cyclin B1 expression and promoted apoptosis and cleaved-caspase-3 expression of two ESCA cell lines [47]. The in-depth study of data on *NOLC1* gene expression and its role in carcinogenesis shows that the protein encoded by this gene is a promising therapeutic target.

Cell life is a result of the interaction and coordinated activity of many proteins working at the same time. Different proteins from different pathways work together to provide a meaningful biological process essential to cell development [47,48,49]. In cancer, protein–protein interaction is essential to form complexes, allowing uncontrolled cellular division, development, growth and tumor promotion. Recently, cancer, neurodegenerative diseases and even infections have been determined to be a result of aberrant protein–protein interactions [49]. In cancer, aberrant proteins interact with other proteins, establishing the initial stages of cancer [48,49]. Based on that, protein–protein interaction analysis has become an alternative target for developing new anticancer compounds [48]. Based on that, a protein–protein interaction analysis of the targets used in this study was performed (Figure 6). The protein–protein interaction analysis revealed that all targets involved in this work are pivotal in their pathways.

For example, the *AJUBA* protein revealed an interaction with the proteins CTNNB1 and β-catenin (Figure 6A). CTNNB1 regulates cellular adhesion and gene transcription during mitotic fuse establishment [50]. The β-catenin protein is a component of the centrosome during the interphase [50]. Bahmanyar et al. [50] suggested that aberrant interactions between the *AJUBA* protein and CTNNB1 and β-catenin lead to uncontrolled cellular division and, thus, cancer development. Our results revealed that the molecule MCULE-2386589557-0-6 interacts with *AJUBA*, leading to a misplaced interaction between *AJUBA* and CTNNB1, interfering in this pathway and probably preventing its role in cancer establishment.

It is important to notice that although our analysis has provided a pipeline to find potential genes that could be used as targets for new anticancer molecules, it is relevant to look to the other side. Our pipeline also provides an analysis of the genes that could be selected as biomarkers of GC prognosis (Figure 4). Despite being either upregulated or downregulated in GC cell lines, genes such as *AJUBA*, *GPNMB*, *CD80*, *PDILT*, *CCDC69*, *KNL1* and *NOLC1* were shown by our results as indicative of a low survival rate for patients (Figure 4). The data revealed that our pipeline has the potential to be employed in both types of studies related to cancer.

## 5. Conclusions

Although tumor markers for different types of cancer have been rapidly discovered in recent years, there is still a lack of specific and sensitive tumor markers for GC management. Combining a systematic collection of public microarray data with a comparative meta-profile approach provided a suitable platform for identifying tumor markers. This activation of pathways shows that these genes are important in carcinogenesis and are probably the result of the convergence of several transforming processes in various cellular contexts. Furthermore, the significant differential expression of these genes implies that they may be useful as therapeutic targets and biomarkers.

Targeted therapies stand out as instruments with great potential for therapeutic success. Our results point to the *AJUBA*, *CD80* and *NOLC1* proteins as candidates for targeted therapy in treating GC. Using a virtual screening approach, a molecular docking study was performed for proteins encoded by genes that play important roles in cellular functions for carcinogenesis.

This study contributed to identifying three compounds with favorable pharmacokinetic, pharmacodynamic and toxicological properties that showed promising results in the molecular docking assay against protein targets, providing information that may help these compounds become potential chemotherapeutics in the clinical therapy of GC. Additional in vivo or in vitro studies may be required to confirm the anticancer activity of these compounds.

## Figures and Tables

**Figure 1 pharmaceutics-15-02303-f001:**
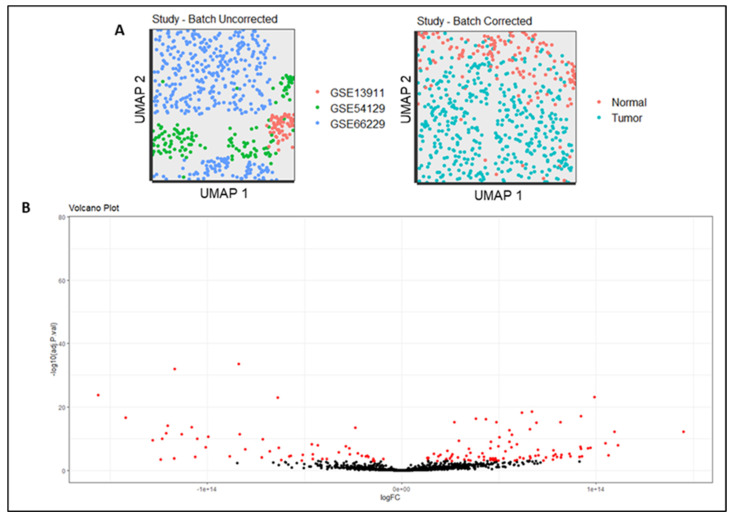
(**A**) Batch effect corrected among the studies. (**B**) Volcano plot of differentially expressed genes (DEGs) in gastric tumors. Red dots represent significant expressed genes.

**Figure 2 pharmaceutics-15-02303-f002:**
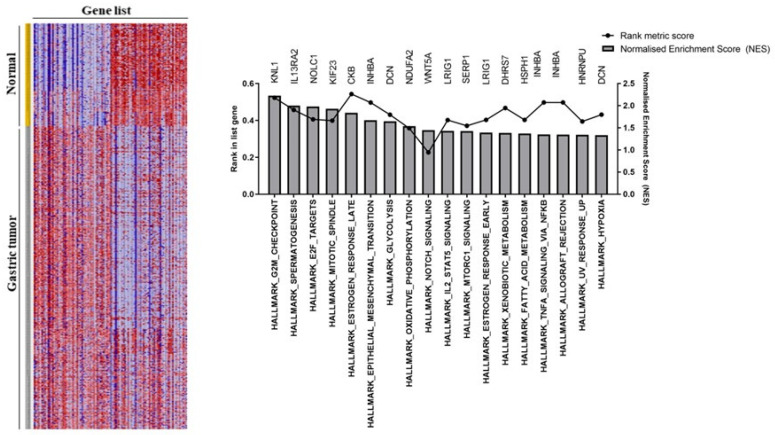
Differentially expressed genes were found using Gene Set Enrichment Analysis (GSEA).

**Figure 3 pharmaceutics-15-02303-f003:**
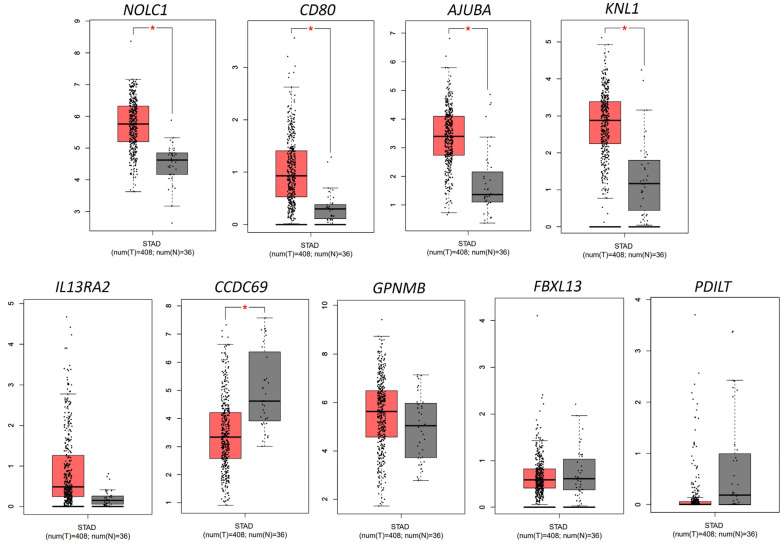
TCGA (RNA-seq) expression data for selected genes using Gene Expression Profiling Interactive Analysis (GEPIA). Bars indicate the difference between treatments. * *p* < 0.01 by one-away (ANOVA) followed by Bonferroni’s test.

**Figure 4 pharmaceutics-15-02303-f004:**
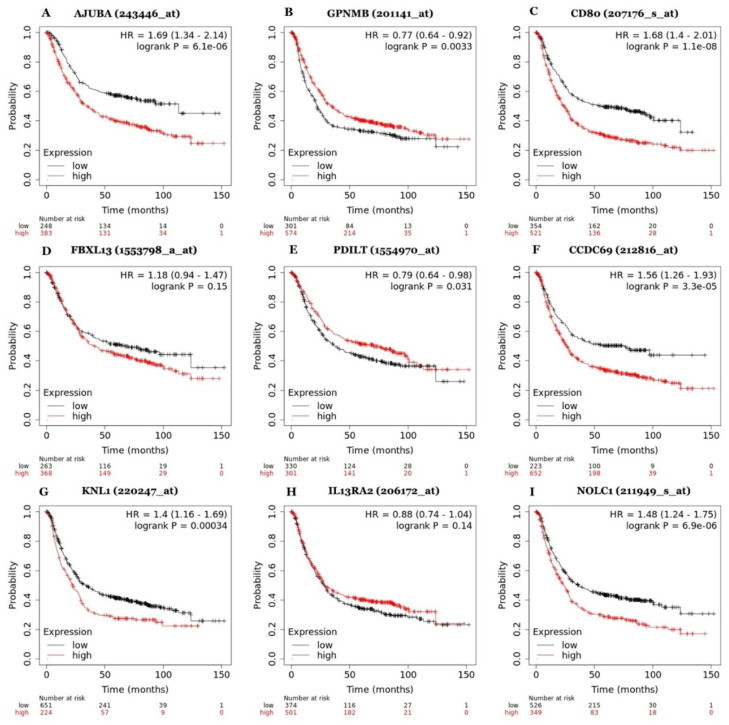
Survival analysis for the genes (**A**) *AJUBA*, (**B**) GPNMB, (**C**) *CD80*, (**D**) FBXL13, (**E**) PDILT, (**F**) CCDC69, (**G**) KNL1, (**H**) IL13RA2 and (**I**) *NOLC1* using Kaplan–Meier plotter (KM plotter).

**Figure 5 pharmaceutics-15-02303-f005:**
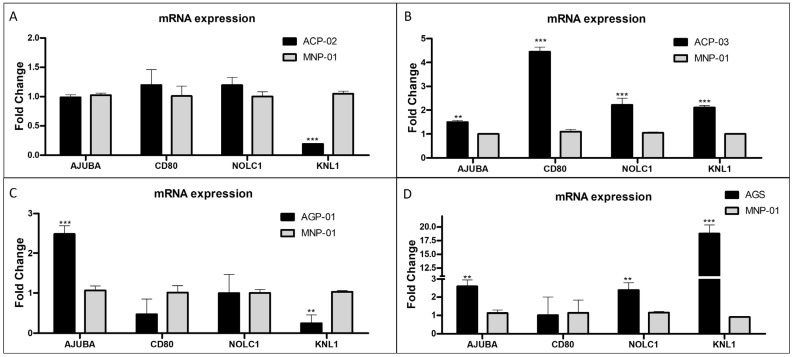
The relative expression level of *AJUBA*, *CD80*, *NOLC1* and KNL1 genes in samples of gastric tumor cell lines (**A**) ACP-02, (**B**) ACP-03, (**C**) AGP-01 and (**D**) AGS is related to normal gastric cells, MNP-01. ** *p* < 0.01; *** *p* < 0.001 by one-away (ANOVA) followed by Bonferroni´s test.

**Figure 6 pharmaceutics-15-02303-f006:**
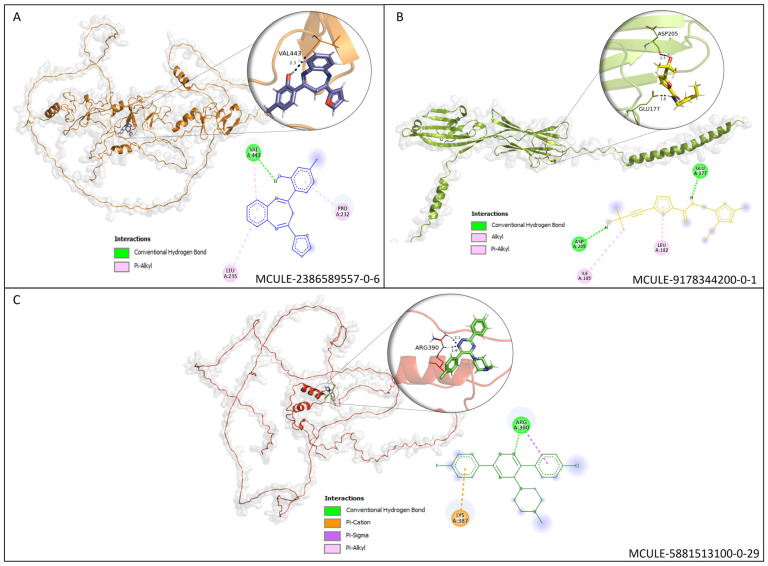
(**A**) Molecular docking analysis of the *AJUBA* protein and molecule MCULE-2386589557-0-6. (**B**) Molecular docking analysis of the *CD80* protein and molecule MCULE-9178344200-0-1. (**C**) Molecular docking analysis of the *NOLC1* protein and molecule MCULE-5881513100-0-29. Figure 6 shows the 2D and 3D analysis of the interaction. In the 3D representation, the three compounds are in stick format; proteins are represented in cartoon format, while the residues involved in hydrogen bonding are shown in line format. Black dotted lines represent hydrogen bonds; the bond length is expressed in the Angstrom (Å) measurement unit. The 2D diagram shows the amino acid residues involved in the interactions between *AJUBA*, *CD80* and *NOLC1* proteins and the selected MCULE compound for each protein. The present type of interaction is represented by colors, which are identified by the legend in the figure.

**Figure 7 pharmaceutics-15-02303-f007:**
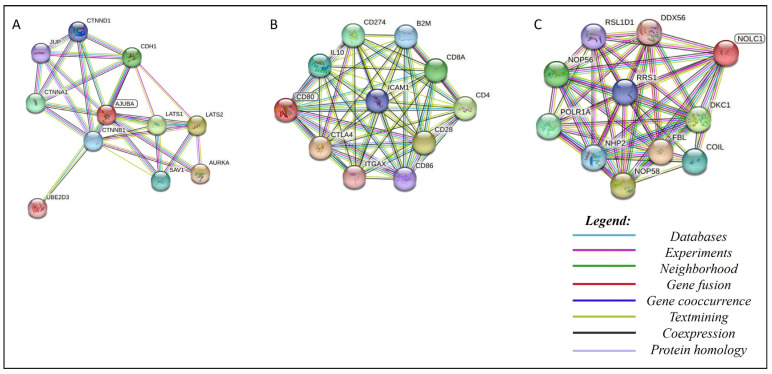
Graphic representation of the network of protein interactions for proteins (**A**) *AJUBA*, (**B**) *CD80* and (**C**) *NOLC1*, identified using STRING v11.5. Each node represents a protein, and each edge represents an interaction. Colored lines between the proteins indicate the various types of interaction evidence.

**Figure 8 pharmaceutics-15-02303-f008:**
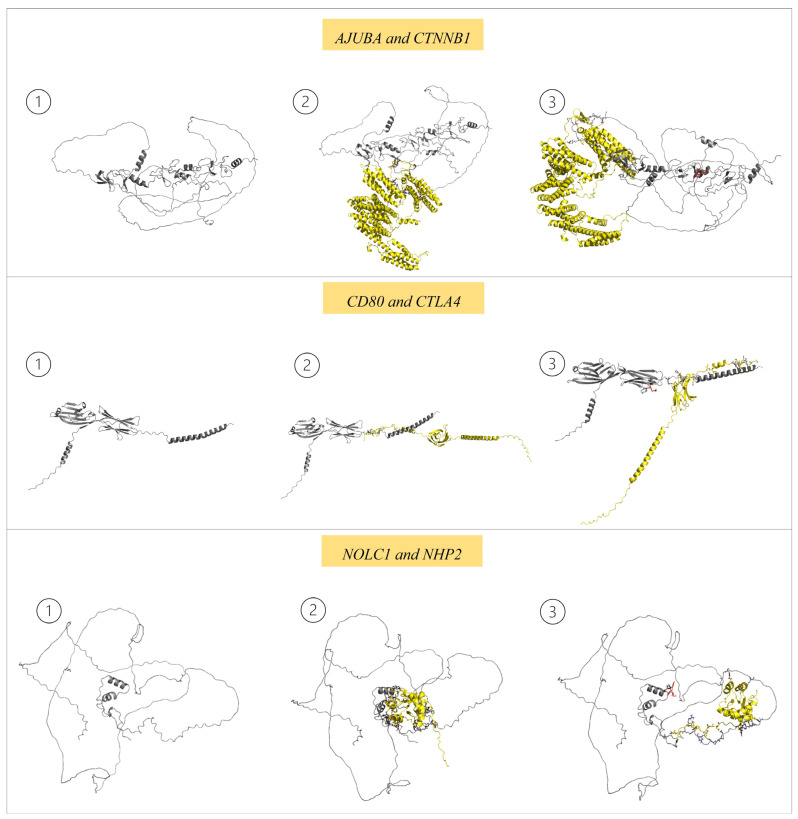
① Protein; ② interaction between the targeted protein and the protein in the respective pathway selected on STRING; ③ analysis of complex formed by the molecule and targeted protein with the protein in the respective pathway.

**Table 1 pharmaceutics-15-02303-t001:** Top 10 up- and downregulated genes in gastric tumors.

Differentially Expressed Genes							
Upregulated				Downregulated			
Gene Symbol	LogFC	t	Adjusted *p*-Value	Gene Symbol	LogFC	t	Adjusted *p*-Value
* AJUBA *	1.44 × 10^14^	7.80	7.78 × 10^13^	FBXL13	−1.56 × 10^14^	−11.18	2.01 × 10^−24^
GPNMB	1.11 × 10^14^	6.20	1.53 × 10^8^	PDILT	−1.14 × 10^14^	−9.20	2.98 × 10^−17^
* CD80 *	1.09 × 10^14^	7.84	5.74 × 10^−13^	CCDC69	−1.28 × 10^14^	−6.83	3.98 × 10^−10^
ANLN	1.06 × 10^14^	4.82	1.61 × 10^−5^	PDZK1IP1	−1.24 × 10^14^	−4.03	4.02 × 10^−14^
ADGRG7	1.06 × 10^14^	4.75	2.22 × 10^−5^	SCIN	−1.23 × 10^14^	−7.05	1.02 × 10^−10^
BICD1	1.04 × 10^14^	6.47	3.44 × 10^−9^	ITIH5	−1.21 × 10^14^	−7.66	1.98 × 10^−12^
KNL1	9.91 × 10^14^	11.01	8.37 × 10^−5^	NKX2-3	−1.20 × 10^14^	−8.46	7.58 × 10^−15^
ABCD3	9.71 × 10^13^	5.89	8.40 × 10^−8^	ITGB1	−1.17 × 10^14^	−4.27	1.59 × 10^−4^
CENPL	9.54 × 10^13^	5.85	1.03 × 10^−7^	SIGLEC11	1.16 × 10^14^	−13.16	1.33 × 10^−32^
PGTS2	9.26 × 10^14^	4.60	4.17 × 10^−5^	PTCHD1	−1.13 × 10^14^	−7.52	5.07 × 10^−12^

Legend: LogFC: fold change in logarithmic scale; t: Student’s t-statistic; P.Value: *p*-value; Adj.P.Val.: adjusted *p*-value.

**Table 2 pharmaceutics-15-02303-t002:** The formulae for the three main compounds. Chemical structure and molecular weight were virtually tested against the protein targets *AJUBA*, *CD80* and *NOLC1*. ACD/ChemSketch (version 2020.2.1) was used to draw the 2D chemical structures of compounds.

Ligand ID	Target	Chemical Formula	Mol. Wt.	2D Structure
1. MCULE-2386589557-0-6	*AJUBA*	C20H16N2O2	316.352	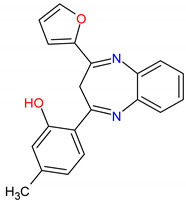
2. MCULE-9178344200-0-1	*CD80*	C17H20N2O3S	332.419	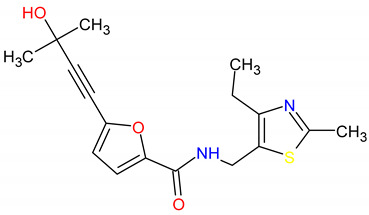
3. MCULE-5881513100-0-29	*NOLC1*	C20H19ClFN5	383.848	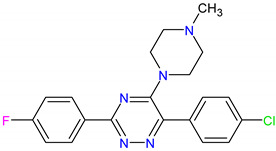

**Table 3 pharmaceutics-15-02303-t003:** Result of docking scores. Using DockThor and Mcule toxicity analysis. Using the ProTox-II tool.

Protein–Ligand Complex		Docking Score		Predicted Toxicity	
Ligand ID	Target	DockThor	Mcule	LD50	Class
MCULE-2386589557-0-6	*AJUBA*	−8.4	−7.3	2500 mg/kg	V
MCULE-9178344200-0-1	*CD80*	−7.7	−5.6	600 mg/kg	IV
MCULE-5881513100-0-29	*NOLC1*	−7.2	−7.5	640 mg/kg	IV

**Table 4 pharmaceutics-15-02303-t004:** In silico prediction of ADME properties of compounds (1) MCULE-2386589557-0-6, (2) MCULE-9178344200-0-1 and (3) MCULE-5881513100-0-29 estimated by SwissADME.

ADME Property	(1)	(2)	(3)
Molecular weight	316.35	332.42	383.85
TPSA	58.09	103.60	45.15
cLogP	3.81	2.73	3.72
LogS	−4.75	−3.49	−4.73
GI absorption	High	High	High
BBB permeant	Yes	No	Yes
P-gp substrate	No	No	Yes

## Data Availability

Data are available under reasonable requirements.

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
