# Peer review of "A Shortcut from Genome to Drug: The Employment of Bioinformatic Tools to Find New Targets for Gastric Cancer Treatment"

_pharmaceutics, 2023, doi:10.3390/pharmaceutics15092303_

Round 1
Reviewer 1 Report
Your research on gastric cancer is impressive, particularly your use of a multi-faceted approach to identify potential target genes and corresponding proteins for the development of new anticancer molecules.
1. Clearly describe the criteria you used to select the three datasets from the Gene Expression Omnibus (GEO) for your analysis. Provide details about the samples, experimental conditions, and any preprocessing steps you performed on the microarray data.
2. It needs Experiment the gain-of-function and loss-of-function experiments for the genes AJUBA (CTNNB1), FBXL13 (SKP1), CCDC69 (NUDT8), CD80 (CTLA4), and NOLC1(NHP2) by RT-qPCR to evaluate gene expression changes in several gastric cancer cell lines or immunoblot to confirm protein level alterations.
3. Explain the rationale behind selecting the molecules MCULE-2386589557-0-6, MCULE-7343047040-0-1, MCULE-5230409338-0-3, MCULE-9178344200-0-1, and MCULE-5881513100-0-29 for your virtual screening. Also, provide information about the DockThor server and the parameters used in the docking analysis.
Overall, your research appears to be a comprehensive and innovative approach to identifying potential therapeutic targets and molecules for gastric cancer treatment. By addressing the points mentioned above, you can further enhance the clarity, depth, and impact of your manuscript.
1. Please check 'CG' in line 15.
2. In line 23, "Our results of transcriptomic analysis, together with RT-qPCR, confirm the high expression of the genes AJUBA, FBXL13, CCDC69, CD80 and NOLC1 in GC lines," consider adding the word "revealed" or "demonstrated" for clarity, like this: "Our results of transcriptomic analysis, together with RT-qPCR, revealed the high expression of the genes AJUBA, FBXL13, CCDC69, CD80, and NOLC1 in GC lines."
Overall, your paper is well-structured and informative. These minor language and style adjustments will further enhance its readability and clarity.
Author Response
Authors´ Response to Reviewer #1
Reviewers' comments:
Reviewer #1 - Comments to the Author
Your research on gastric cancer is impressive, particularly your use of a multi-faceted approach to identify potential target genes and corresponding proteins for the development of new anticancer molecules.
Authors’ Response
Dear Reviewer #1
We are thankful for you expending time to review our manuscript. Certainly, your suggestion will bring the manuscript to a higher scientific level.
Reviewer #1 – Comment #1
- Clearly describe the criteria you used to select the three datasets from the Gene Expression Omnibus (GEO) for your analysis. Provide details about the samples, experimental conditions, and any preprocessing steps you performed on the microarray data.
Authors’ Response #1
Dear Reviewer #1, Thank you for that comment. However, we kindly ask you to look again at the manuscript because that information are already there. You can find them at Pg 3. Section 2.1 Lns 99 to 105. Or you can see it below.
Reviewer #1 – Comment #2
- It needs Experiment the gain-of-function and loss-of-function experiments for the genes AJUBA (CTNNB1), FBXL13 (SKP1), CCDC69 (NUDT8), CD80 (CTLA4), and NOLC1(NHP2) by RT-qPCR to evaluate gene expression changes in several gastric cancer cell lines or immunoblot to confirm protein level alterations.
Authors’ Response #2
Thank you for that comment. We understand your concern and are grateful for this comment. We are planning to do that in second work, which is about to start. At this moment, we sought to look for target and find potential molecules.
Reviewer #1 – Comment #3
- Explain the rationale behind selecting the molecules MCULE-2386589557-0-6, MCULE-7343047040-0-1, MCULE-5230409338-0-3, MCULE-9178344200-0-1, and MCULE-5881513100-0-29 for your virtual screening. Also, provide information about the DockThor server and the parameters used in the docking analysis.
Authors’ Response #3
Dear reviewer 1,
You are completely right. To explain better why those molecules where chosen, the Supplementary Tables 3-5 have new information added. Now those tables have information about toxicity endpoint models, the data of DockThor as requested by you. Additionally, we provide a punctuation score following methodology of Amaral et al. 2021 to clarify how those molecules were selected.
Reviewer #1 – Comment #4
Overall, your research appears to be a comprehensive and innovative approach to identifying potential therapeutic targets and molecules for gastric cancer treatment. By addressing the points mentioned above, you can further enhance the clarity, depth, and impact of your manuscript.
Authors’ Response #3
Dear reviewer 1,
Thank you for your suggestions and accomplishments about our manuscript. We are glad about that. We are sure that you comment improve and bring our manuscript to a higher level.
reviewer #1 minor comments
- Please check 'CG' in line 15.
- In line 23, "Our results of transcriptomic analysis, together with RT-qPCR, confirm the high expression of the genes AJUBA, FBXL13, CCDC69, CD80 and NOLC1 in GC lines," consider adding the word "revealed" or "demonstrated" for clarity, like this: "Our results of transcriptomic analysis, together with RT-qPCR, revealed the high expression of the genes AJUBA, FBXL13, CCDC69, CD80, and NOLC1 in GC lines."
Overall, your paper is well-structured and informative. These minor language and style adjustments will further enhance its readability and clarity.
Authors’ Response
All of them were fixed. Thank you.

Reviewer 2 Report
This manuscript proposed a bioinformatic pipeline to in silico identify potential small molecule drugs for gastric cancer. The tools used in this pipeline include toxicity prediction by ProTox-II, ADME prediction by SwissADME, molecular docking by DockThor, PPI prediction by STRING, and re-docking by ClusPro. This research set up a useful example of small molecule virtual screening specific to gastric cancer. The presentation is clear and the methods are solid.
Author Response
Authors´ Response to Reviewer #2
Reviewer #2 Comments to the Author
This manuscript proposed a bioinformatic pipeline to in silico identify potential small molecule drugs for gastric cancer. The tools used in this pipeline include toxicity prediction by ProTox-II, ADME prediction by SwissADME, molecular docking by DockThor, PPI prediction by STRING, and re-docking by ClusPro. This research set up a useful example of small molecule virtual screening specific to gastric cancer. The presentation is clear and the methods are solid.
Authors’ General Response
Dear Reviewer #2
We are thankful for you expending time to review our manuscript. We are glad for the comments you have made on our manuscript. Thank you so much.

Reviewer 3 Report
Brito et al predicted new targets for gastric cancer treatment it is good study.
I have a following concerns.
1) Authors used microarray datasets to analyze the gene expression data but it is very old technology. I suggest to validate the genes expression using TCGA bulk RNA-seq data and proteome datasets of gastric cancer. It will provide robust analysis
2) 3D structures of proteins are obtained from Alpha Fold database which used AI system but it is still computational models. I suggest validation of these protein structures using Ramachandran plots and see all the torsion angels are in favored region or not.
3) In results section 3.1 authors selected only top 3 genes (up and down) is there any particular reason for this selection criterion?
4) In Figure 3 FBXL13 and IL13RA2 did not show significant survival curves then why FBXL13 was studies further in docking and other analysis?
5) In section 3.8 authors conducted PPI analysis it is not clear why this is done?
Author Response
Authors´ Response to Reviewer #3
Reviewer #3 – Comment to the Author
Brito et al predicted new targets for gastric cancer treatment it is good study.
I have the following concerns.
Authors’ General Response
Dear reviewer #3
We are thankful for you expending time to review our manuscript and for your comment.
Reviewer #3 – Comment #1
1) Authors used microarray datasets to analyze the gene expression data but it is very old technology. I suggest to validate the genes expression using TCGA bulk RNA-seq data and proteome datasets of gastric cancer. It will provide robust analysis
Authors’ Response #1
Dear reviewer #3
As suggested by you, we have used the TCGA dataset to validate our analysis and results are in consonance. The new analysis generated new figures and results are described in the new version of the manuscript. Thank you for that comment.
Reviewer #3 – Comment #2
2) 3D structures of proteins are obtained from Alpha Fold database which used AI system but it is still computational models. I suggest validation of these protein structures using Ramachandran plots and see all the torsion angels are in favored region or not.
Authors’ Response #2
Dear reviewer #3
Sorry for that mistake. All the models have more than 85% of quality on alphafold and were checked by two servers Verify 3D and ERRATA. This information was added in the methodology.
Reviewer #3 – Comment #2
2) 3D structures of proteins are obtained from Alpha Fold database which used AI system but it is still computational models. I suggest validation of these protein structures using Ramachandran plots and see all the torsion angels are in favored region or not.
Authors’ Response #2
Dear reviewer #3
Sorry for that mistake. All the models have more than 85% of quality on alphafold and were checked by two servers Verify 3D and ERRATA. This information was added in the methodology.
Reviewer #3 – Comment #3
3) In results section 3.1 authors selected only top 3 genes (up and down) is there any particular reason for this selection criterion?
Authors’ Response #3
Dear reviewer #3
Sorry but we think the criteria was clear. The explanation was added in the text of the new version of the manuscript.
Reviewer #3 – Comment #4
4) In Figure 3 FBXL13 and IL13RA2 did not show significant survival curves then why FBXL13 was studies further in docking and other analysis?
Authors’ Response #4
Dear reviewer #3
You are completely right. Based on your comment we did the analysis again and removed those genes and respective proteins. Thank you.
Reviewer #3 – Comment #5
5) In section 3.8 authors conducted PPI analysis it is not clear why this is done?
Authors’ Response #5
Dear reviewer #3
The PPI analysis was done to provide an idea about how our molecules could affect the pathway of the target. In addition, the PPI allowed us to find targets for redocking analysis. We have reviewed the text in the manuscript to clarify it.
